# High-Temperature Characterization of AlGaN Channel High Electron Mobility Transistor Based on Silicon Substrate

**DOI:** 10.3390/mi15111343

**Published:** 2024-10-31

**Authors:** Yinhe Wu, Xingchi Ma, Longyang Yu, Xin Feng, Shenglei Zhao, Weihang Zhang, Jincheng Zhang, Yue Hao

**Affiliations:** 1Guangzhou Wide Bandgap Semiconductor Innovation Center, Guangzhou Institute of Technology, Xidian University, Guangzhou 510555, China; wuyinhe@xidian.edu.cn (Y.W.); 23111213596@stu.xidian.edu.cn (X.M.); xinfeng@xidian.edu.cn (X.F.); whzhang@xidian.edu.cn (W.Z.); 2State Key Laboratory of Wide-Bandgap Semiconductor Devices and Integrated Technology, School of Microelectronics, Xidian University, Xi’an 710071, China; slzhao@xidian.edu.cn (S.Z.); jchzhang@xidian.edu.cn (J.Z.); yhao@xidian.edu.cn (Y.H.)

**Keywords:** AlGaN channel high electron mobility transistor, high temperature, trap states, carrier mobility

## Abstract

In this paper, it is demonstrated that the AlGaN high electron mobility transistor (HEMT) based on silicon wafer exhibits excellent high-temperature performance. First, the output characteristics show that the ratio of on-resistance (*R*_ON_) only reaches 1.55 when the working temperature increases from 25 °C to 150 °C. This increase in *R*_ON_ is caused by a reduction in optical phonon scattering-limited mobility (*μ*_OP_) in the AlGaN material. Moreover, the device also displays great high-performance stability in that the variation of the threshold voltage (Δ*V*_TH_) is only 0.1 V, and the off-state leakage current (*I*_D,off-state_) is simply increased from 2.87 × 10^−5^ to 1.85 × 10^−4^ mA/mm, under the operating temperature variation from 25 °C to 200 °C. It is found that the two trap states are induced at high temperatures, and the trap state densities (*D_T_*) of 4.09 × 10^12^~5.95 × 10^12^ and 7.58 × 10^12^~1.53 × 10^13^ cm^−2^ eV^−1^ are located at *E*_T_ in a range of 0.46~0.48 eV and 0.57~0.61 eV, respectively, which lead to the slight performance degeneration of AlGaN HEMT. Therefore, this work provides experimental and theoretical evidence of AlGaN HEMT for high-temperature applications, pushing the development of ultra-wide gap semiconductors greatly.

## 1. Introduction

Owing to the high electric field strength and electron mobility of the GaN material, GaN HEMTs have been widely researched and applied in power electronics [1,2,3,4,5]. Compared with traditional GaN materials, ultra-wide bandgap (UWBD) materials possess higher bandgaps and critical electric fields; thus, these devices based on UWBD materials have more developmental potential in power electronic applications [6,7,8,9,10,11]. Among these UWBD materials (β-Ga_2_O_3_, diamond, and AlN), the material growth and device fabrication of AlGaN appear to be much easier due to the relevant GaN-based heterojunction. Moreover, AlGaN HEMTs have been recognized as an excellent candidate for high-temperature applications because of their great thermal stability [12,13]. The outstanding performance of AlGaN HEMTs under extreme temperature conditions makes them an ideal choice for a wide range of applications, including aerospace, electric vehicles, radar and communication, energy conversion, and industrial automation.

For these reasons, many scholars have carried out extensive studies on the device characteristics of AlGaN HEMTs at high temperatures. Numerous valuable achievements have been obtained, including lower drain current attenuation, off-state leakage current, on-resistance degradation, channel mobility attenuation, smaller threshold voltage shift, higher breakdown electric field, saturated drain current, and current gain cut-off frequency of AlGaN channel HEMTs compared with traditional GaN channel HEMTs [14,15,16,17,18,19,20,21,22,23,24,25]. However, most of these reports are based on sapphire and AlN substrates, and there is a lack of research on the performance and related analysis of the AlGaN HEMTs based on silicon (Si-based AlGaN HEMTs) at high temperatures. As is known, the Si-based AlGaN HEMTs have better thermal conductivity and greater commercialization potential due to the use of low-cost substrates. Therefore, it is necessary to investigate the high-temperature characteristics of Si-based AlGaN HEMTs and identify the relevant influence factors for performance degeneration so as to put forward more useful strategies to solve it.

In this work, the technology of pulsed metal–organic chemical vapor deposition (MOCVD) was used to fabricate the 6-inch AlGaN wafer on the Si substrate. Then, the Si-based AlGaN HEMT with 400 nm Al_0.1_Ga_0.9_N as a channel layer was fabricated and showed a low contact resistance of 0.95 Ω·mm and a high breakdown voltage of 1110 V (with *L*_GD_ = 15 μm). In addition, the output and transfer characteristics of this device at high-temperature working conditions were systematically investigated. Using the alternating current conductance method, the trapping effect in the AlGaN channel was characterized by time constants (*τ*_T_) and *D*_T_, which helped to explain the performance degradation. Overall, given the cost advantages of Si-based AlGaN HEMTs, this study provides valuable references for future design optimizations of such devices and also lays the theoretical foundation for their practical applications in harsh environments, such as those involving high temperatures and high powers.

## 2. Materials and Methods

The pulsed MOCVD system was used to grow AlGaN material on a 6-inch Si substrate. Firstly, an AlN nucleation layer was deposited. Then, a 3 μm graded Al_x_Ga_1−x_N layer, 1 μm C-doped Al_0.1_Ga_0.9_N layer, 400 nm Al_0.1_Ga_0.9_N channel layer, 1 nm AlN insertion layer, 24 nm Al_0.4_Ga_0.6_N barrier layer, and 2 nm GaN cap layer were grown above the AlN nucleation layer as a sequence. The HRXRD rocking curves from (0002) and (10–12) planes of the AlGaN channel layer in this work are 708 and 1147 arcsec, respectively.

The fabrication process of AlGaN HEMTs began with mesa isolation etching. Thereafter, a 30 nm recess was etched accurately using the inductively coupled plasma (ICP) technique, and the Ti/Al alloy was deposited and annealed to form ohmic contact as a source/drain electrode. The Ni/Al alloy was deposited as a gate electrode. Finally, 40 nm Al_2_O_3_ was grown on the surface of AlGaN HEMTs by plasma-enhanced atomic layer deposition (PEALD) to suppress the effect of surface states. The device tested in this work had a gate width (W_G_) of 100 µm, a gate length (*L**_G_***) of 4 µm, a gate–source distance (*L**_GS_***) of 4 µm, and a gate–drain distance (*L**_GD_***) of 15 µm. The cross-sectional structure of AlGaN HEMT is shown in Figure 1. The output and transfer characteristics were tested using the Agilent B1505A high-voltage semiconductor analyzer system (Keysight Technologies, Beijing, China).

## 3. Results and Discussion

As shown in Figure 2a, the DC output characteristics of Si-based AlGaN HEMTs were measured at 25~200 °C with a growing step of 25 °C under a given *V*_G_ of 3 V. At 25 °C, the maximum output current (*I*_D,max_) of the AlGaN HEMTs is 323.14 mA/mm, displaying a gradually decreasing trend as the temperature rises. The *I*_D,max_ still maintains values of 212.04 and 189.50 mA/mm at high temperatures of 150 and 200 °C, respectively. So, the ratio *R*_ON,150 °C_/*R*_ON,25 °C_ for the Si-based AlGaN HEMT is only 1.55 in this work, which is significantly lower than the corresponding values of commercial GaN HEMTs on silicon (the values of *R*_ON,150 °C_/*R*_ON,25 °C_ of the GaN systems GS 66516B and GS 66502B are about 2.58). Therefore, the Si-based AlGaN HEMT demonstrates more stable output characteristics compared with GaN HEMTs at high-temperature applications [9,10].

The degeneration of *I*_D,max_ at high temperatures may be caused by an increase in ohmic contact resistance (*R*_C_) and 2DEG channel resistance (*R*_CH_). *R*_ON_ can be realized as *R*_ON_ = *R*_S_ + *R*_D_ + 2*R*_C_ + *R*_CH_. *R*_S_ or *R*_D_ is the parasitic source/drain resistance. Usually, *R*_S_ and *R*_D_ are much lower than *R*_C_. As shown in Figure 2b, *R*_C_ is calculated using a transmission line model (TLM) at different temperatures. The variation of *R*_C_ is very poor with the growing temperature. Compared with the variation of *R*_ON_, the variation of *R*_C_ is negligible. However, it is observed that the sheet resistance (*R*_SH_) increases from 947.0 Ω/□ to 1945.7 Ω/□ as the temperature rises from 25 °C to 200 °C, indicating that the rise in R_ON_ can be attributed to increased channel resistance, which can be interpreted as decreased carrier mobility.

The temperature-dependent C-V characteristics of the Schottky barrier diode with an AlGaN channel are shown in Figure 3a. The electron concentration of the AlGaN channel varied with the temperature-dependent depth profiles and are derived by the following Equations (1) and (2) [26,27], and the relevant results are shown in Figure 3b:*Z* = *εA*/*C*(1)
(2)$$n_{c}=-\frac{2}{e\varepsilon A^{2}}\times \frac{1}{d\left(1/c^{2}\right)/dV}q$$
(3)$$n_{2DEG}=\int_{-\infty }^{+\infty }n_{c}\left(Z\right)dZ$$
μ = 1/(R_SH_ × n_s_ × q) (4)
where *Z* is the depth to the gate, *ε* is the dielectric constant of the barrier, *C* is the junction capacitor, and *A* is the area of the junction capacitor. The dielectric constant used in this work is $\varepsilon_{r,Al_{0.4}Ga_{0.6}N}=8.66$. The *n*_2*DEG*_ is approximately calculated by Equation (3). As shown in Figure 3c, with a temperature increase, the variation of *n*_2*DEG*_ is very poor.

As shown in Figure 3d, the carrier mobility of the AlGaN channel (*μ*_AlGaN_) is approximately calculated according to Equation (4) [28], which decreases from 691.58 to 339.47 cm^2^/v·s as the temperature increases from 25 °C to 200 °C, and it is in good agreement with the simulation result obtained by alloy scattering-limited and optical phonon-limited mobility models [13]. As is known, the *μ*_AlGaN_ is affected jointly by alloy disorder scattering and optical phonon scattering, where *μ*_OP_ is very sensitive to temperature and alloy scattering-limited mobility (*μ*_alloy_) is inverse [13]. When the temperature rises, the influence of *μ*_OP_ is gradually reduced while part of *μ*_alloy_ still remains stable, leading to the idea that the AlGaN HEMTs have very excellent output characteristics under high-temperature conditions, something which is lacking in the GaN HEMTs because *μ*_OP_ is very dominating in terms of carrier mobility.

Furthermore, the transfer characteristics of AlGaN HEMTs were measured at 25~200 °C with a step of 25 °C under a *V*_D_ of 10 V and are shown in Figure 4a. *V*_TH_ shifts slightly towards a positive direction, from −4.6 V to −4.5 V, as the temperature increases. *I*_D,off-state_ also increases marginally from 2.87 × 10^−5^ to 1.85 × 10^−4^ mA/mm. Additionally, the buffer leakage current (*I*_buffer_) measured with an isolated active-area mesa structure is only 9.5 × 10^−8^ mA/mm at 200 °C, which is significantly lower than *I*_D,off-state_. Therefore, an increase in *I*_D,off-state_ at high temperatures is attributed to the reverse gate leakage current, as shown in Figure 4b. And it can be suppressed by optimizing the barrier layer or adopting a metal–insulator–semiconductor (MIS) structure.

The AC capacitance technique was used to characterize the trapping effect on AlGaN HEMTs at a room temperature of 25 °C and a high temperature of 200 °C. The frequency ranged from 1 kHz to 300 kHz. As shown in Figure 5a,b, the conductance gradually increases and then decreases with the growing radial frequency. It is observed that only one conductance peak exists for a given *V*_G_ at 25 °C in Figure 5a, which can be assumed as only one kind of trap state existing in the AlGaN channel. However, there are two conductance peaks occurring at 200 °C in Figure 5b, indicating that two kinds of trap states exist in the AlGaN channel under high-temperature conditions.

Here, the plots in Figure 5a,b are effectively fitted as these curves by the following equations, respectively:(5)$$\frac{G_{p}}{\omega }=\frac{q\omega \tau_{T}D_{T}}{1+{\left(\omega \tau_{T}\right)}^{2}}$$
(6)$$\frac{G_{p}}{\omega }=\frac{q\omega \tau_{T,1}D_{T,1}}{1+{\left(\omega \tau_{T,1}\right)}^{2}}+\frac{q\omega \tau_{T,2}D_{T,2}}{1+{\left(\omega \tau_{T,2}\right)}^{2}}$$

Therefore, the time constant (*τ*_T_) and trap *D*_T_ can be extracted from the above equations. The relationship between *τ*_T_ and the applied gate voltage is depicted in Figure 5c at 25 °C and 200 °C, respectively, where *τ*_T_ is distributed in the range of 1.08~3.13 μs for the device working on 25 °C but two types of values of 0.53~0.77 μs and 6.29~17.2 μs for that on 200 °C, indicating the existence of two kinds of trap states at the same time. So, these states can be calculated by the equation as follows:(7)τT=σTNcvT−1expETkT

Here, the cross-section of the trap states ($\sigma_{T}$) is captured as 3.4 × 10^−15^ cm^−2^, and the density of states in the conduction band (*N*_c_) is 2.2 × 10^18^ cm^−3^. The average thermal velocity of carriers (*v_T_*) is 2.6 × 10^7^ and 1.69 × 10^7^ cm/s for 25 and 200 °C, respectively. As shown in Figure 5d, *D*_T_ from the range of 4.43 × 10^12^~6.67 × 10^12^ cm^−2^ eV^−1^ is located at *E**_T_*** in a range of 0.32~0.35 eV at 25 °C. But when the temperature increases to 200 °C, *D*_T_ with a distribution of 4.09 × 10^12^~5.95 × 10^12^ and 7.58 × 10^12^~1.53 × 10^13^ cm^−2^ eV^−1^ is obviously located at *E**_T_*** in the range of 0.46~0.48 eV and 0.57~0.61 eV, respectively. Compared with the reported trapping effects in AlGaN materials on sapphire by Zhao’s work [29], the *τ_T_* and *D*_T_ of trap states are much lower in the AlGaN material on silicon, indicating that the AlGaN epilayer material quality has improved significantly.

## 4. Conclusions

In summary, the high-temperature characteristics of the Si-based AlGaN HEMTs are studied sufficiently in this work. The carrier mobility of the AlGaN channel still remains at 339.47 cm^2^/v·s when the working temperature reaches 200 °C, and *I*_D,max_ still reaches up to 189.50 mA/mm. Moreover, only a positive voltage drift of 0.1 V occurs, and *I*_D,off-state_ increases marginally from 2.87 × 10^−5^ to 1.85 × 10^−4^ mA/mm, strongly proving the outstanding working performance of Si-based AlGaN HEMTs compared with GaN at high temperatures. Finally, the inner trapping effect in Si-based AlGaN materials is investigated using the AC capacitance technique. Two types of trap states are found at a high temperature of 200 °C. The first type of trap state has a *D*_T_ ranging from 4.09 × 10^12^ to 5.95 × 10^12^ cm^−2^ eV^−1^ and is located within an *E_T_* of 0.46 to 0.48 eV. The second type of trap state has a *D*_T_ ranging from 7.58 × 10^12^ to 1.53 × 10^13^ cm^−2^ eV^−1^ and is located within an *E_T_* of 0.57 to 0.61 eV. Hence, this work is very valuable for Si-based AlGaN materials as the channel for HEMTs, especially for high-temperature applications, and significantly pushes the development of ultra-wide bandgap semiconductors in power electronics.

## Figures and Tables

**Figure 1 micromachines-15-01343-f001:**
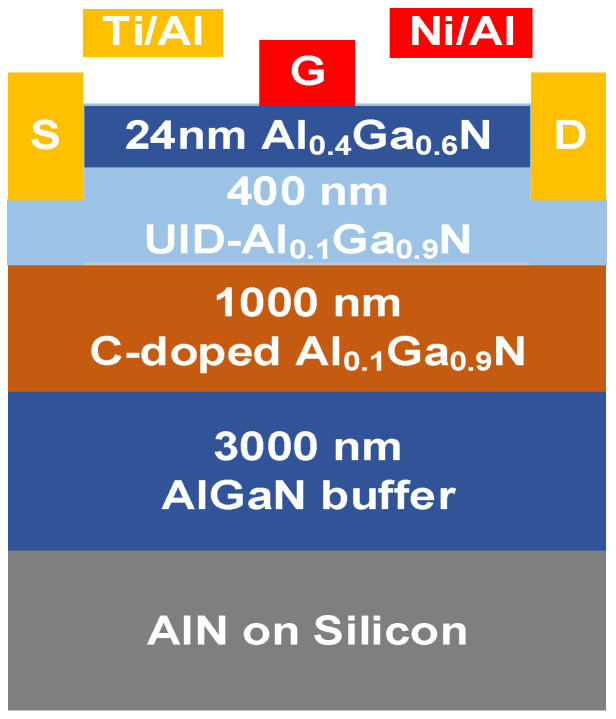
The cross-sectional structure of Si-based AlGaN HEMT.

**Figure 2 micromachines-15-01343-f002:**
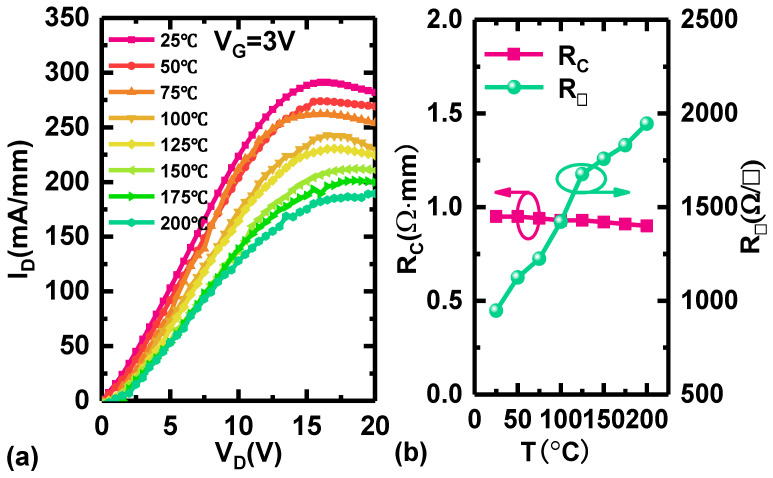
(**a**) The output characteristics of Si-based AlGaN HEMT at 25/50/75/100/125/150/175/200 °C. (**b**) The variation of *R*_C_ and *R*_SH_ at different temperatures.

**Figure 3 micromachines-15-01343-f003:**
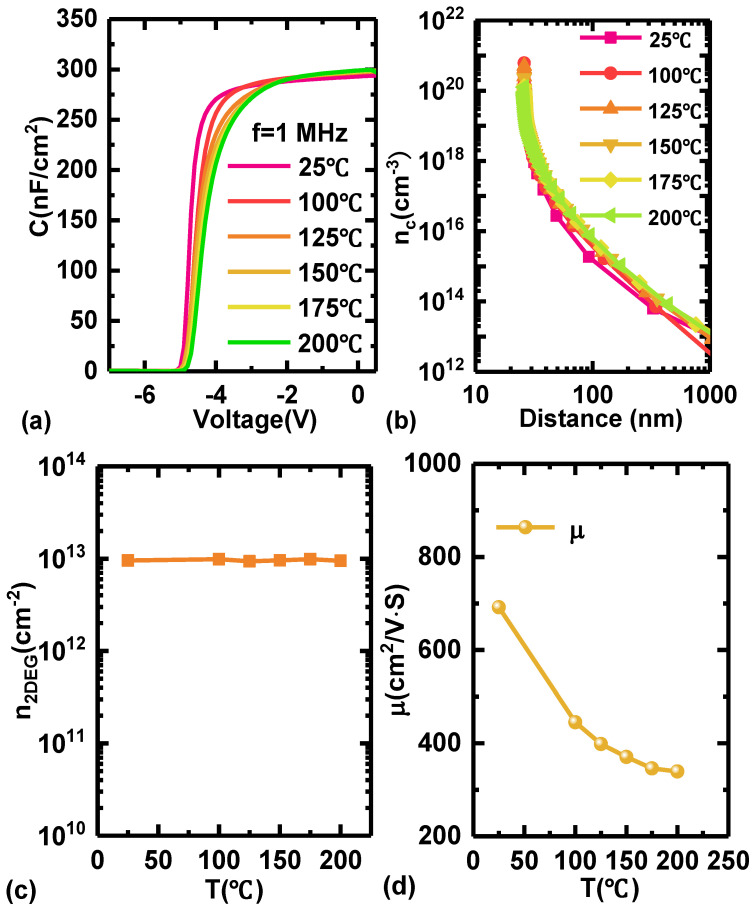
(**a**) The temperature-dependent C-V characteristics of the Schottky barrier diode on the AlGaN channel. (**b**) The value of carrier concentration with the variation of channel depth. (**c**) The temperature-dependent 2DEG sheet density (*n*_2*DEG*_). (**d**) The variation of carrier mobility in the AlGaN channel with the increasing temperature.

**Figure 4 micromachines-15-01343-f004:**
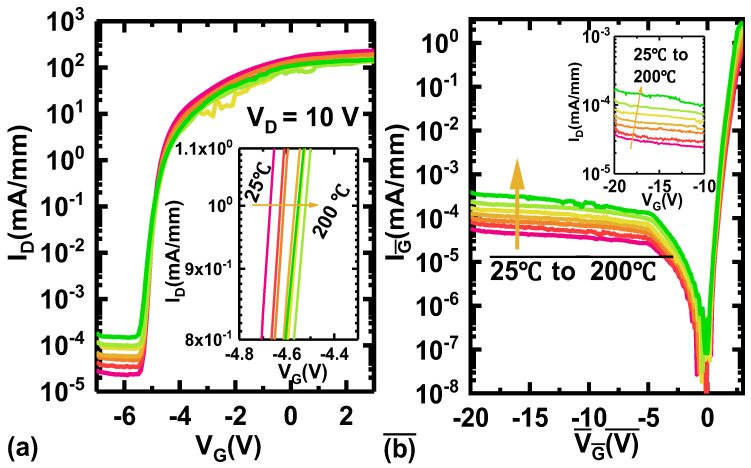
(**a**) The transfer characteristics of Si-based AlGaN HEMTs at 25/75/100/125/150/175/200 °C with an applied *V*_D_ of 10 V. (**b**) The forward and reverse I-V characteristics of the Schottky barrier diode in the AlGaN channel at different temperatures.

**Figure 5 micromachines-15-01343-f005:**
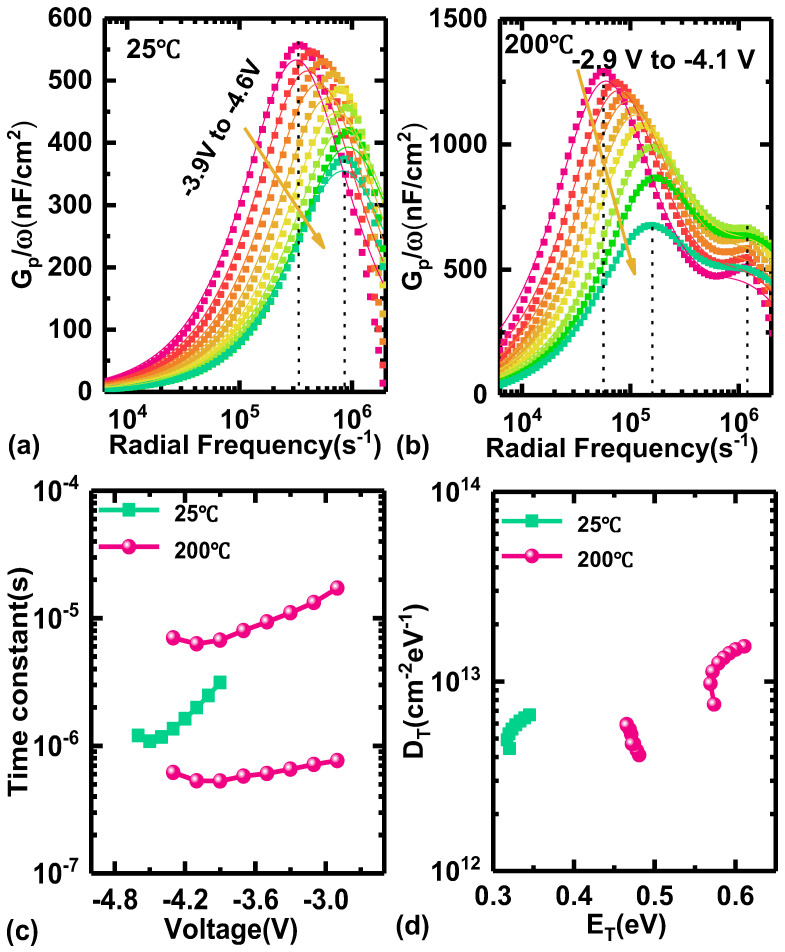
Conductance as a function of radial frequency for Si-based AlGaN HEMTs at (**a**) 25 °C and (**b**) 200 °C. (**c**) The time constant of trap states as a function of gate voltage. (**d**) The trap state density as a function of their energy.

## Data Availability

The original contributions presented in the study are included in the article, further inquiries can be directed to the corresponding author.

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
