# Peer review of "High-Temperature Characterization of AlGaN Channel High Electron Mobility Transistor Based on Silicon Substrate"

_micromachines, 2024, doi:10.3390/mi15111343_

Round 1
Reviewer 1 Report
Comments and Suggestions for Authors
High amount of work is given in the manuscript, including AlGaN HEMT fabrication, characterisation. This manuscript could be improved:
Introduction should be improved by highlighting the problem or missing knowledge. It would be nice to get some overview in which applications AlGaN HEMTS are used at such elevated temperatures and why this study is required. The novelty and significance of the research should be clearly pointed out.
There is not much novelty in the paper except two traps at elevated temperatures. Those traps could be deeper investigated by using pulsed IV capabilities of the B1505A. This could help to extract trap relaxation time.
If the traps are existing in the system, what about drain current hysteresis. Which current is presented in Fig.2 and Fig.4? I assume that forward? what about reverse current?
Conclusions are weak. There is too little novelty if authors target power electronics applications. The sentence (170-174) is too long that it is difficult to understand. Split the long sentences to short and clear statements.
How to understand the statement: "...that traps are leading to an optical phonon scattering..."? Explain in details what is the relation between slow traps and very fast optical phonons?
Comments on the Quality of English Language
Check English language with spelling or verification tools. Some sentences are not complete or too long and very difficult to understand what was done or meant (Lines-14-16), (39-41), (52-54).
There are strange symbols (subscripts) (Line 91, Fig.2,b).
Reviewer 2 Report
Comments and Suggestions for Authors
In this work, the high-temperature characteristics for Si-based AlGaN HEMT are studied. Here the carrier mobility of AlGaN channel still keep as 339.47 cm2/v·s when the working temperature is going to 200 ℃, and ID, max still could reach up to 189.50 mA/mm, proving the outstanding working performance of Si-based AlGaN HEMT than that of GaN at high temperature. The inner trapping effect in Si-based AlGaN material is investigated according to the AC-Capacitance technique, where two kind of trap states are found at high temperature of 200℃, leading to the optical phonon scattering of carriers. Overall, this work may be desirable for Si-based AlGaN material as the channel for HEMTs, especially for the high-temperature application, and pushing hugely the development of ultrawide gap semiconductor in power electronics. However, there are many big flaws in the manuscript. Major revisions are needed before it can be published.
1) Abstract: “Here, it’s demonstrated that the AlGaN…” should be revised as “Here, it is demonstrated that the AlGaN…”. As a scientific paper, the sentences should be full. Please carefully check whole manuscript to exclude such kind of errors.
2) Key word: The full name of HETM should be provided in the key word. Moreover, “High temperature characterization” is not suitable to be used as a key word.
3) INTRODUCTION part: This part is too short. The novelty of this study should be highlighted. What are the differences between previous reports and the current study? What is the novelty of this work? These two aspects should be clearly presented in the introduction part. As a scientific paper, only 16 references are involved, more literatures should be cited. Meanwhile, the literatures cited are too old, which cannot reflect the latest advances in the research field. I noted that the latest literatures cited published in 2020.
4) I don’t think that it’s a qualified scientific paper in the current form. It’s very rough, and numerous improvements are need before it can be accepted.
Comments on the Quality of English LanguageThere arr numerous gramatical errors.
Round 2
Reviewer 2 Report
Comments and Suggestions for Authors
can be accepted